# The incidence, characteristics and outcomes of pregnant women hospitalized with symptomatic and asymptomatic SARS-CoV-2 infection in the UK from March to September 2020: A national cohort study using the UK Obstetric Surveillance System (UKOSS)

Nicola Vousden[1,2], Kathryn Bunch[1], Edward Morris[3], Nigel Simpson[4], Christopher Gale[5], Patrick O'Brien[6], Maria Quigley[1], Peter Brocklehurst[7], Jennifer J. Kurinczuk[1], Marian Knight[1] *

**1** National Perinatal Epidemiology Unit, Nuffield Department of Population Health, University of Oxford, Oxford, United Kingdom, **2** School of Population Health & Environmental Sciences, King's College London, London, United Kingdom, **3** Royal College of Obstetricians and Gynaecologists, London, United Kingdom, **4** Department of Women's and Children's Health, School of Medicine, University of Leeds, Leeds, United Kingdom, **5** Neonatal Medicine, School of Public Health, Faculty of Medicine, Imperial College London, London, United Kingdom, **6** Institute for Women's Health, University College London, London, United Kingdom, **7** Birmingham Clinical Trials Unit, Institute of Applied Health Research, University of Birmingham, Birmingham, United Kingdom

* marian.knight@npeu.ox.ac.uk

## Abstract

### Background

There is a lack of population level data on risk factors, incidence and impact of SARS-CoV-2 infection in pregnant women and their babies. The primary aim of this study was to describe the incidence, characteristics and outcomes of hospitalized pregnant women with symptomatic and asymptomatic SARS-CoV-2 in the UK compared to pregnant women without SARS-CoV-2.

### Methods and findings

We conducted a national, prospective cohort study of all hospitalized pregnant women with confirmed SARS-CoV-2 from 01/03/2020 to 31/08/2020 using the UK Obstetric Surveillance System. Incidence rates were estimated using national maternity data. Overall, 1148 hospitalized women had confirmed SARS-CoV-2 in pregnancy, 63% of which were symptomatic. The estimated incidence of hospitalization with symptomatic SARS-CoV-2 was 2.0 per 1000 maternities (95% CI 1.9–2.2) and for asymptomatic SARS-CoV-2 was 1.2 per 1000 maternities (95% CI 1.1–1.4). Compared to pregnant women without SARS-CoV-2, women hospitalized with symptomatic SARS-CoV-2 were more likely to be overweight or obese (adjusted OR 1.86, (95% CI 1.39–2.48) and aOR 2.07 (1.53–2.29)), to be of Black, Asian or Other minority ethnic group (aOR 6.24, (3.93–9.90), aOR 4.36, (3.19–5.95) and aOR 12.95,

**Data Availability Statement:** Data cannot be shared publicly because of confidentiality issues and potential identifiability of sensitive data as identified within the Research Ethics Committee application/approval. Requests to access the data can be made by contacting the National Perinatal Epidemiology Unit data access committee via general@npeu.ox.ac.uk.

**Funding:** The study was funded by the National Institute for Health Research HS&DR Programme (project number 11/46/12; https://www.nihr.ac.uk/explore-nihr/funding-programmes/health-services-and-delivery-research.htm). MK is an NIHR Senior Investigator who received this award. MK, MQ, PB, PO'B, JJK received grants from the NIHR in relation to the submitted work. The views expressed are those of the authors and not necessarily those of the NHS, the NIHR or the Department of Health and Social Care. The funder played no role in study design; data collection and analysis, decision to publish or preparation of the manuscript. The corresponding author (MK) had full access to all the data in the study and had final responsibility for the decision to submit for publication.

**Competing interests:** MK, MQ, PB, PO'B, JJK received grants from the NIHR in relation to the submitted work. KB, NV, NS, CG have no conflicts of interest to declare. EM is Trustee of RCOG, British Menopause Society and Newly Chair of the Board of Trustees Group B Strep Support. This does not alter our adherence to PLOS ONE policies on sharing data and materials.

(4.93–34.01)), and to have a relevant medical comorbidity (aOR 1.83 (1.32–2.54)). Hospitalized pregnant women with symptomatic SARS-CoV-2 were more likely to be admitted to intensive care (aOR 57.67, (7.80–426.70)) but the absolute risk of poor outcomes was low. Cesarean births and neonatal unit admission were increased regardless of symptom status (symptomatic aOR 2.60, (1.97–3.42) and aOR 3.08, (1.99–4.77); asymptomatic aOR 2.02, (1.52–2.70) and aOR 1.84, (1.12–3.03)). The risks of stillbirth or neonatal death were not significantly increased, regardless of symptom status.

## Conclusions

We have identified factors that increase the risk of symptomatic and asymptomatic SARS-CoV-2 in pregnancy. Clinicians can be reassured that the majority of women do not experience severe complications of SARS-CoV-2 in pregnancy.

## Introduction

In March 2020 the World Health Organization declared a global pandemic of novel coronavirus infection (SARS-CoV-2) [1]. Evidence about risk factors, incidence and impact of SARS-CoV-2 infection in pregnant women and their babies has rapidly expanded and is vital to planning guidance and policy. The World Health Organization's (WHO) living systematic review concluded that SARS-CoV-2 infection was associated with increased risk of admission to intensive care for the woman and increased risk of preterm birth and admission to neonatal care for the infant [2]. Women with pre-existing medical comorbidities, older age, high body mass index (BMI) and women of Black, Asian and minority ethnic groups have been reported to be at increased risk of hospitalization [3] or severe outcome [2]. However, the majority of studies to date are case reports, case series and institutional or registry non-population-based cohort studies and there is a lack of population-level data to inform accurate incidence rates and unbiased descriptions of characteristics and outcomes.

Clinical practice around testing for SARS-CoV-2 among pregnant women in the UK has changed since the start of the pandemic, when predominantly only those with symptoms were tested. Routine screening of all obstetric admissions was recommended by the Royal College of Obstetricians and Gynaecologists (RCOG) on 29th May 2020 [4] and therefore the UK's obstetric population is unique in that virtually all were tested thereafter, typically at the time of giving birth. The WHO systematic review reported from 11 small observational studies (n = 162 women that had universal screening) and suggested that a high proportion of women who tested positive were asymptomatic (74%, 95% CI 51%-93%) [2]. Other reports have varied [5, 6], with between 79% (from a total of 55 women) [7] to 100% (from a total of 17 women) [8] of those that tested positive on universal screening being asymptomatic. However, these studies were small and were undertaken in single hospitals or regions. No published studies to date have explored the proportion of symptomatic versus asymptomatic SARS-CoV-2 in pregnancy at the population level since universal screening was introduced.

The primary aim of this study was to describe the incidence, characteristics and outcomes of hospitalized pregnant women with symptomatic and asymptomatic SARS-CoV-2 in the UK compared to women without SARS-CoV-2, in order to inform ongoing guidance and management. The second aim was to describe characteristics and outcomes in women with symptomatic SARS-CoV-2 compared to those who remained asymptomatic.

## Materials and methods

This on-going national, prospective observational cohort study was conducted using the UK Obstetric Surveillance System (UKOSS) [9]. UKOSS is a research platform that was established in 2005 to collect national population-based information about specific severe pregnancy complications. All 194 hospitals in the UK with a consultant-led maternity unit participate, and thus the mechanism to collect comprehensive information about women hospitalized with SARS-CoV-2 in pregnancy was already in place at the start of the pandemic. Nominated reporting clinicians were asked to notify all pregnant women admitted to their hospital with confirmed SARS-CoV-2. The process was enabled by research midwives and nurses from the UK's National Institute of Health Research Clinical Research Network following the study's adoption as an urgent public health priority study [10]. To check for completeness, a monthly reporting email was sent in addition to receipt of live reports, and zero reports were confirmed. Following notification of a case, clinicians completed an electronic data collection form containing anonymized details of women's demographics, management and birth and perinatal outcomes. Reporters who had notified a case but not returned data were contacted by email at one, two and three weeks after notification. This analysis reports characteristics and outcomes of women who were hospitalized from 1st March 2020 to 31st August 2020. Hospital admission was defined as a hospital stay of 24 hours or longer for any cause, or admission of any duration to give birth. Women were taken as confirmed SARS-CoV-2 if they were hospitalized during pregnancy or within two days of giving birth and had a positive test during or within seven days of admission, or they were symptomatic and had evidence of pneumonia on imaging which was typical of SARS-CoV-2. Women were excluded if they did not meet this case definition (n = 294).

For each woman included, characteristics were described: body mass index, age, ethnicity, pregnancy history and relevant pre-existing comorbidities, which were identified based on current NHS guidance (S1 Table) [11]. Women's ethnic groups were identified on maternal self-report as recorded in the maternity records, and were classified based on the census classification for England and Wales (S2 Table) [12]. Details of pregnancy outcome including admission to intensive care, evidence of pneumonia on imaging, pre-eclampsia and mode and indication for birth were described as well as infant outcomes including gestation at birth, stillbirth, live birth, admission to neonatal intensive care, neonatal death and neonatal testing for SARS-CoV-2. Information on women who died, or who had consequent stillbirths or neonatal deaths, was cross-checked with data from the MBRRACE-UK collaboration (Mothers and Babies: Reducing Risk through Audits and Confidential Enquiries across the UK), the organization responsible for maternal and perinatal death surveillance in the UK [13]. If any women were identified through these sources who had not been identified for this study, the nominated UKOSS clinician in the relevant hospital was contacted and asked to complete a data collection form.

Women with any symptoms of SARS-CoV-2 (fever, cough, sore throat, breathlessness, headache, fatigue, limb or joint pain, vomiting, rhinorrhea, diarrhea, anosmia, or SARS CoV-2 pneumonia on imaging) that were admitted to hospital were compared to a historical comparison cohort of uninfected women. The historical comparison cohort were obtained from a previous study of seasonal influenza in pregnancy, where the two women giving birth immediately prior to any woman hospitalized with confirmed influenza between 1st November 2017 and 30th October 2018 were identified [14]. A historical cohort was used to ensure there was no possibility that comparison women had asymptomatic or minimally symptomatic SARS-CoV-2 infection. Women who tested positive during routine screening at the time of hospital admission with no symptoms at any point were also compared to this historical

comparison group. A sub-analysis compared women admitted to hospital with symptomatic SARS-CoV-2 and women found to have asymptomatic SARS-CoV-2.

### Study registration

The study is registered with ISRCTN, number 40092247, and is still open to case notification. The study protocol is available at https://www.npeu.ox.ac.uk/ukoss/current-surveillance/covid-19-in-pregnancy.

### Role of the funding source

The funder played no role in study design; in the collection, analysis, and interpretation of data; in the writing of the report; nor the decision to submit the paper for publication. The corresponding author (MK) had full access to all the data in the study and had final responsibility for the decision to submit for publication.

### Ethics and consent

This study was approved by the HRA NRES Committee East Midlands–Nottingham 1 (Ref. Number: 12/EM/0365). Data cannot be shared publicly because of confidentiality issues and potential identifiability of sensitive data as identified within the Research Ethics Committee application/approval. Requests to access the data can be made by contacting the National Perinatal Epidemiology Unit data access committee via general@npeu.ox.ac.uk.

### Statistical methods and analysis

Statistical analyses were performed using STATA version 15 (Statacorp, TX, USA). Numbers and proportions are presented with 95% confidence intervals. Where data were missing, proportions are presented out of cases known. Odds ratios (ORs) with 95% confidence intervals (CI) were estimated using unconditional logistic regression. Exploratory analysis, using a hierarchical stepwise forward selection procedure to add additional variables associated with the outcome in the univariable analysis with subsequent likelihood ratio testing, was used to identify any potential confounders or mediators for each analysis (P-value <0.05 considered significant for inclusion in the model). Continuous variables were used in this exploratory analysis then categorized. In addition, maternal age was included as a potential confounder as identified in a previous preliminary unpublished analysis [15] and therefore was included in the multivariate model for the comparison of symptomatic SARS-CoV-2 to the historical comparison group. Any further potential confounders identified as significantly associated during the univariable analysis were adjusted in the multivariable unconditional regression analysis. Plausible interactions were tested by the addition of interaction terms and subsequent likelihood ratio testing on removal, with a P-value <0.01 considered as evidence of significant interaction to account for multiple testing. Variance inflation factors were examined for evidence of significant multicollinearity.

A sensitivity analysis was undertaken to explore if any pre-existing medical problems (e.g., diabetes), that might have increased the risk of infection with SARS-CoV-2 or resulting morbidity, were independently associated with the outcome. In this national observational study, the study sample size was governed by the disease incidence, thus no formal power calculation was carried out. The most recently available (2018) national maternity data for the constituent countries of the United Kingdom was used as the denominator to estimate the incidence of hospitalization with confirmed SARS-CoV-2 infection in pregnancy.

## Results

Between 1st March 2020 and 31st August 2020 there was a total of 1148 hospitalized women with confirmed SARS-CoV-2 infection in pregnancy in the UK, approximately two thirds of whom were symptomatic (n = 722, 63%). There were an estimated 364,830 maternities during this period, giving an overall incidence of confirmed SARS-CoV-2 in women hospitalized in pregnancy of 3.1 per 1000 maternities (95% CI 3.0–3.3), an incidence of symptomatic SARS-CoV-2 of 2.0 per 1000 maternities (95% CI 1.9–2.2) and an incidence of asymptomatic SAR-CoV-2 of 1.2 per 1000 maternities (95% CI 1.1–1.4). Most cases were in the first month of the pandemic as shown in Fig 1, which also demonstrates the greater proportion of symptomatic compared to asymptomatic cases at this time. The majority (99%, n = 1136) had SARS-CoV-2 confirmed on laboratory testing and 12 symptomatic women (1%) were diagnosed on imaging alone.

The most common time of diagnosis was during the third trimester (n = 355, 50% of symptomatic women and n = 211, 50% of asymptomatic women). The primary reason for hospital admission was known in 177 of 291 women admitted after universal screening was recommended. The most common reason for asymptomatic women to be admitted to hospital was to give birth (68%, n = 78), whereas the primary reason that symptomatic women were admitted to hospital during the same period was equally spread between admission for SARS-CoV-2, to give birth, and for other reasons (30%, n = 19, 37%, n = 23 and 33% n = 21 respectively). In symptomatic women, the majority had experienced symptoms within two weeks of admission to hospital (n = 645, 94% of 689 where date of onset was known).

Compared to the historical comparison cohort without SARS-CoV-2, those hospitalized with symptomatic SARS-CoV-2 were more likely to be overweight or obese (33% vs. 27%, adjusted OR 1.86, 95% CI 1.39–2.48 and 34% vs. 23% aOR 2.07, 95% CI 1.53–2.29 respectively) (Table 1). More than half (55%, n = 391) of women with symptomatic SARS-CoV-2 were from Black, Asian or Other minority ethnic groups, compared with 19% (n = 131) of the historical comparison cohort and the odds of admission for these groups were significantly increased (Black ethnicity aOR 6.24, 95% CI 3.93–9.90, Asian ethnicity aOR 4.36, 95% CI 3.19–5.95 and

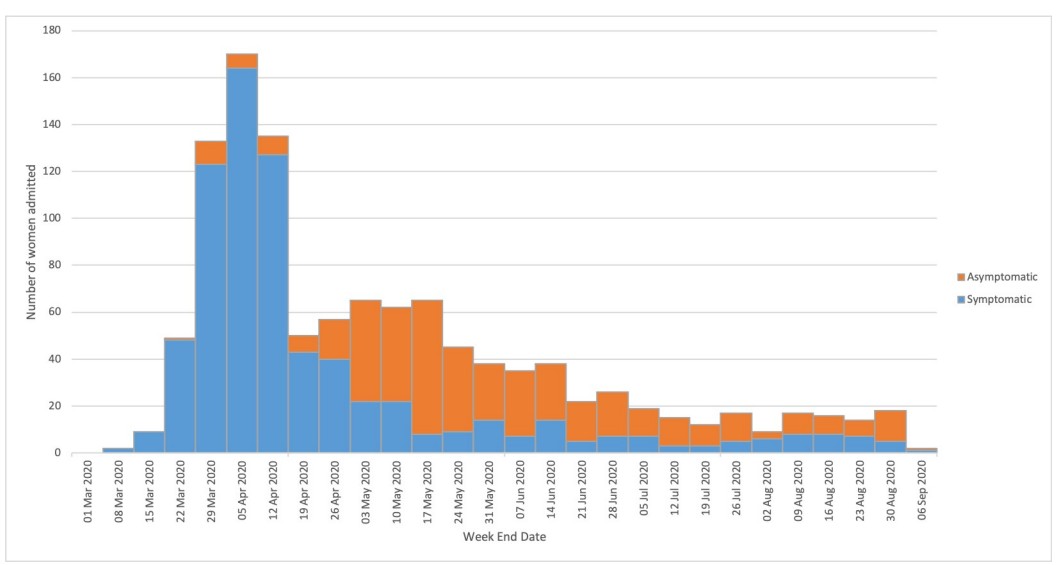

**Fig 1. Number of pregnant women admitted to hospital with symptomatic and asymptomatic confirmed SARS-CoV-2 infection in the UK between 1st March and 31st August 2020.**

**Table 1. Characteristics of pregnant women with symptomatic confirmed SARS-CoV-2 infection admitted to hospital in the UK compared to a historical cohort without SARS-CoV-2 infection.**

| Characteristic | Women with symptomatic SARS-CoV-2 (N = 722) | Historical comparison cohort (N = 694) | OR (95% CI) | aOR** |
|---|---|---|---|---|
| | | | P-value | P-value |
| | Number (%) * | Number (%) * | | |
| **Age (years):** | | | | |
| <20 | 12 (2%) | **18 (3%)** | 0.71 (0.34–1.48) | **1.38 (0.57–3.35)** |
| | | | p = 0.356 | **p = 0.472** |
| **20–34** | 451 (62%) | **477(69%)** | 1 | **1** |
| ≥35 | 258 (36%) | **199 (29%)** | 1.37 (1.09–1.72) | **1.09 (0.84–1.42)** |
| | | | P = 0.006 | **p = 0.512** |
| **Missing** | 12 (2%) | **0** | | |
| **Body Mass index (BMI):** | | | | |
| **Normal** | 221 (32%) | **337 (50%)** | 1 | **1** |
| **Overweight** | 237 (33%) | **181 (27%)** | 2.00 (1.54–2.58) | **1.86 (1.39–2.48)** |
| | | | p<0.001 | **p<0.001** |
| **Obese** | 235 (34%) | **155 (23%)** | 2.31 (1.77–3.01) | **2.07 (1.53–2.79)** |
| | | | p<0.001 | **p<0.001** |
| **Missing** | 27 | **18** | - | - |
| **Either woman or partner in paid work** | 574 (80%) | **537 (77%)** | 1.13 (0.88–1.46) | Omitted |
| | | | p = 0.331 | |
| **Ethnic Group** | | | | |
| **White** | 318 (45%) | **558 (81%)** | 1 | **1** |
| **Asian** | 210 (30%) | **79 (11%)** | 4.66 (3.48–6.25) | **4.36 (3.19–5.95)** |
| | | | p<0.001 | **p<0.001** |
| **Black** | 122 (17%) | **26 (4%)** | 8.23 (5.28–12.85) | **6.24 (3.93–9.90)** |
| | | | p<0.001 | **p<0.001** |
| **Chinese** | 8 (1%) | **7 (1%)** | 2.01 (0.72–5.58) | **1.93 (0.68–5.52)** |
| | | | p = 0.183 | **P = 0.219** |
| **Other** | 36 (5%) | **5 (1%)** | 12.63 (4.91–32.52) | **12.95 (4.93–34.01)** |
| | | | p<0.001 | **p<0.001** |
| **Mixed** | 15 (2%) | **14 (2%)** | 1.88 (0.90–3.95) | **1.72 (0.78–3.80)** |
| | | | P = 0.095 | **p = 0.183** |
| **Missing** | 13 | **5** | - | - |
| **Current smoking** | 42 (6%) | **135 (20%)** | 0.26 (0.18–0.38) | **0.42 (0.28–0.62)** |
| | | | p<0.001 | **p<0.001** |
| **Missing** | 35 | **10** | - | - |
| **Any relevant pre-existing medical problems** | 156 (22%) | **90 (13%)** | 1.85 (1.39–2.46) | **1.83 (1.32–2.54)** |
| | | | p<0.001 | **p<0.001** |
| **Asthma** | 49 (7%) | **31 (4%)** | 1.56 (0.98–2.47) | - |
| | | | p = 0.060 | |
| **Hypertension** | 24 (3%) | **3 (<1%)** | 7.92 (2.37–26.42) | - |
| | | | p = 0.001 | |
| **Cardiac disease** | 13 (2%) | **10 (1%)** | 1.25 (0.54–2.88) | - |
| | | | p = 0.593 | |
| **Diabetes** | 22 (3%) | **7 (1%)** | 3.08 (1.31–7.27) | - |
| | | | p = 0.010 | |

*(Continued)*

**Table 1.** (Continued)

| Characteristic | Women with symptomatic SARS-CoV-2 (N = 722) | Historical comparison cohort (N = 694) | OR (95% CI) P-value | aOR** P-value |
|---|---|---|---|---|
| **Multiparous** | 436 (60%) | **420 (61%)** | 1.01 (0.81–1.25) p = 0.937 | Omitted |
| Missing | 4 | **0** | - | - |
| **Multiple pregnancy** | 12 (2%) | **13 (2%)** | 0.89 (0.30–1.95) p = 0.763 | Omitted |
| **Gestational diabetes** | 76 (11%) | **37 (5%)** | 2.09 (1.39–3.14) p<0.001 | Omitted |
| **Gestation at diagnosis (weeks)** | | | | |
| <22 | 53 (7%) | | | |
| 22–27 | 66 (9%) | | | |
| 28–31 | 98 (14%) | | | |
| 32–36 | 126 (17%) | | | |
| 37 or more | 131 (18%) | | | |
| Peripartum | 241 (34%) | | | |
| Missing | 7 | | | |

* Percentages of those with complete data.

** adjusted for ethnicity, BMI, any relevant previous medical problem, smoking.

Other minority ethnicity aOR 12.95, 95% CI 4.93–34.01). Nearly a quarter of women (22%, n = 156) admitted with symptomatic SARS-CoV-2 had a relevant medical comorbidity compared to 13% (n = 90) of the historical comparison cohort (aOR 1.83, 95% CI 1.32–2.54). In the sensitivity analysis, there was some evidence that asthma and hypertension specifically increased the risk of admission with symptomatic SARS-CoV-2, although the numbers in some groups were small (aOR 2.12, 95% CI 1.25–3.58 and aOR 3.63, 95% CI 0.99–13.30 respectively, S3 Table). These risk factors were similar when comparing the overall group of women admitted to hospital with SARS-CoV-2 (both symptomatic and asymptomatic) to the historical comparison cohort of women without SARS-CoV-2 (S4 Table).

Women with asymptomatic SARS-CoV-2 on hospital admission were also more likely to be of Black or minority Asian ethnicity compared to the historical comparison cohort (Black ethnicity aOR 2.54, 95% CI 2.48–4.34, Asian ethnicity aOR 2.09, 95% CI 1.48–2.95 and Other ethnicity aOR 6.90, 95% CI 2.47–19.23) (Table 2). Women with asymptomatic SARS-CoV2 were also more likely to have gestational diabetes compared to the historical comparison cohort (aOR 1.68, 95% CI 1.02–2.74), however raised BMI and pre-existing medical co-morbidities were no longer associated (Table 2). Therefore, there were also differences between women with symptomatic SARS-CoV-2 and asymptomatic SARS-CoV-2 as identified in the sub-analysis (S5 Table).

There were eight deaths of hospitalized women with symptomatic SARS-CoV-2 during this period, two of which were unrelated to SARS-CoV-2. This gives a maternal mortality rate of 2.2 hospitalized women per 100,000 maternities (95% CI 0.9–4.3). 63 (5%) women required critical care, with four (<1%) reported to have received extracorporeal membrane oxygenation (ECMO) (Table 3). In those women admitted for critical care there were five maternal deaths (8%); the majority of women admitted to critical care with SARS-CoV-2 were discharged from hospital (n = 56, 92%, S6 Table). Whilst most women admitted to critical care gave birth before 37 weeks of pregnancy (n = 38, 64%), with 32% (n = 17) being born before 32 weeks, short term infant outcomes were good with 98% (n = 60) being liveborn (S7 Table).

**Table 2.** Characteristics of pregnant women with asymptomatic confirmed SARS-CoV-2 infection admitted to hospital in the UK compared to a historical cohort without SARS-CoV-2 infection.

| Characteristic | Women with asymptomatic SARS-CoV-2 (N = 426) | Historical comparison cohort (N = 694) | OR (95% CI) | aOR** |
|---|---|---|---|---|
| | Number (%) * | Number (%) * | | |
| **Age (years):** | | | | |
| <20 | 11 (3%) | 18 (3%) | 0.93 (0.43–2.00) | 1.17 (0.54–2.52) |
| | | | p = 0.855 | P = 0.686 |
| 20–34 | 313 (73%) | 477(69%) | 1 | 1 |
| ≥35 | 102 (24%) | 199 (29%) | 0.78 (0.59–1.03) | 0.70 (0.53–0.94) |
| | | | p = 0.082 | P = 0.018 |
| Missing | 1 | 0 | - | - |
| **Body Mass index (BMI):** | | | | |
| Normal | 188 (46%) | 337 (50%) | 1 | Omitted |
| Overweight | 111 (27%) | 181 (27%) | 1.10 (0.82–1.48) | Omitted |
| | | | p = 0.531 | |
| Obese | 110 (27%) | 155 (23%) | 1.27 (0.94–1.72) | Omitted |
| | | | p = 0.119 | |
| Missing | 16 | 18 | - | |
| **Either woman or partner in paid work** | 323 (76%) | 537 (77%) | 0.92 (0.69–1.22) | Omitted |
| | | | p = 0.549 | |
| **Ethnic Group** | | | | |
| White | 276 (66%) | 558 (81%) | 1 | 1 |
| Asian | 84 (20%) | 79 (11%) | 2.15 (1.53–3.02) | 2.09 (1.48–2.95) |
| | | | P<0.001 | p<0.001 |
| Black | 33 (8%) | 26 (4%) | 2.57 (1.50–4.38) | 2.54 (1.48–4.34) |
| | | | p = 0.001 | p = 0.001 |
| Chinese | 4 (1%) | 7 (1%) | 1.16 (0.34–3.98) | 1.24 (0.36–4.29) |
| | | | p = 0.819 | p = 0.734 |
| Other | 16 (4%) | 5 (1%) | 6.47 (2.35–17.84) | 6.90 (2.47–19.23) |
| | | | P<0.001 | p<0.001 |
| Mixed | 5 (1%) | 14 (2%) | 0.72 (0.26–2.03) | 0.76 (0.27–2.14) |
| | | | p = 0.536 | p = 0.604 |
| Missing | 8 | 5 | - | - |
| **Current smoking** | 57 (16%) | 135 (20%) | 0.79 (0.56–1.10) | Omitted |
| | | | p = 0.165 | |
| Missing | 74 | 10 | - | - |
| **Any relevant pre-existing medical problems** | 64 (15%) | 90 (13%) | 1.19 (0.84–1.68) | Omitted |
| | | | p = 0.333 | |
| Asthma | 28 (7%) | 31 (4%) | 1.50 (0.89–2.55) | - |
| | | | p = 0.128 | |
| Hypertension | 2 (<1%) | 3 (<1%) | 1.08 (0.18–6.53) | - |
| | | | p = 0.928 | |
| Cardiac disease | 8 (2%) | 10 (1%) | 1.31 (0.51–3.34) | - |
| | | | p = 0.573 | |
| Diabetes | 6 (1%) | 7 (1%) | 1.40 (0.47–4.20) | - |
| | | | p = 0.546 | |
| **Multiparous** | 239 (57%) | 420 (61%) | 0.86 (0.67–1.09) | Omitted |
| Missing | 5 | 0 | | - |

*(Continued)*

**Table 2.** (Continued)

| Characteristic | Women with asymptomatic SARS-CoV-2 (N = 426) | Historical comparison cohort (N = 694) | OR (95% CI) | aOR** |
|---|---|---|---|---|
| | Number (%) * | Number (%) * | | |
| **Multiple pregnancy** | 4 (1%) | **13 (2%)** | 0.50 (0.16–1.53) | **Omitted** |
| | | | p = 0.223 | |
| **Gestational diabetes** | 40 (9%) | **37 (5%)** | 1.84 (1.16–2.92) | **1.68 (1.02–2.74)** |
| | | | p = 0.010 | **p<0.001** |
| **Gestation at diagnosis (weeks)** | | | | |
| **<22** | 23 (5%) | | | |
| **22–27** | 11 (3%) | | | |
| **28–31** | 8 (2%) | | | |
| **32–36** | 39 (9%) | | | |
| **37 or more** | 164 (39%) | | | |
| **Peripartum** | 180 (42%) | | | |
| **Missing** | 1 | | | |

* Percentages of those with complete data.

** adjusted for ethnicity, age and gestational diabetes.

Of the 722 women admitted to hospital with symptomatic SARS-CoV-2, 89% (n = 640) had completed their pregnancy at the time of analysis (Table 3, two were missing further details and infant outcomes so were excluded from the denominator). In total, 2% (n = 16) of women with symptomatic SARS-CoV-2 had a pregnancy loss prior to 24 weeks'. Early pregnancy outcomes were not compared with the historical comparison group due to the risk of measurement bias resulting from screening for SARS-CoV-2 on admission to hospital with symptoms of pregnancy loss. Nearly half of women gave birth by cesarean section (n = 314, 49%), with 64 (6%) being for maternal compromise secondary to SARS-CoV-2 (S7 Table). This represents approximately double the risk of cesarean section for women with symptomatic SARS-CoV-2 compared to the historical comparison cohort without SARS-CoV-2 (pre-labor cesarean aOR 2.58, 95% CI 1.88–3.55 and cesarean after labor onset aOR 2.72, 95% CI 1.79–3.86). Operative vaginal births were also increased (aOR 2.14, 95% CI 1.42–3.24). Only 18 women received antiviral agents, two of whom were recruited to the RECOVERY Trial [16]. The most commonly used antiviral agent was oseltamivir (n = 11, 1%). Approximately 1 in 8 symptomatic women received steroids to enhance fetal lung maturation (17%, n = 120) and 1 in 5 symptomatic women had a preterm birth. The majority of preterm births were iatrogenic and the risk of a woman with symptomatic SARS-CoV-2 having an iatrogenic preterm birth was more than 10-fold higher compared to pregnant women without SARS-CoV-2 (14% vs. 1% aOR 11.43, 95% CI 5.07–25.75). Whilst there was an apparently lower proportion of spontaneous preterm births amongst women admitted with symptomatic SARS-CoV-2, this was partly explained by confounding (4% vs. 7%, aOR 0.57, 95% CI 0.32–1.01, Table 3).

In comparison with women found to have asymptomatic SARS-CoV-2 on hospital admission, women with symptomatic SARS-CoV-2 were more likely to have a cesarean birth (prelabor cesarean: 32% vs. 26%, OR 1.51, 95% CI 1.11–2.06 and cesarean after labor onset: 18% vs. 14%, OR 1.58, 95% CI 1.08–2.31) (Table 4). Although the risk in the asymptomatic SARS-CoV-2 group was also increased compared to the historical comparison cohort without SARS-CoV-2 (Table 5) (pre-labor cesarean 26% vs. 18%, aOR 2.26, 95% CI 1.62–3.17; cesarean after labor onset 14% vs 11%, aOR 1.67, 95% CI 1.12–2.52).

**Table 3. Pregnancy and infant outcomes for pregnant women with symptomatic confirmed SARS-CoV-2 infection hospitalized in the UK compared to a historical cohort without SARS-CoV-2 infection.**

| Characteristic | Women with symptomatic SARS-CoV-2 (N = 722) | Historical comparison cohort (N = 694) | OR (95% CI) | aOR (95% CI)** |
|---|---|---|---|---|
| | Number (%) * | Number (%) * | | |
| **Required critical care** | 63 (9) | 1 (<1) | 66.15 (9.15–478.32) | **57.67 (7.80–426.70)** |
| | | | p<0.001 | **p<0.001** |
| **SARS-CoV-2 pneumonia** | 173 (15) | 0 | - | - |
| **Pre-eclampsia** | 15 (2%) | 8 (1) | 1.82 (0.77–4.32) | **1.37 (0.52–3.61)** |
| | | | p = 0.175 | **p = 0.522** |
| **Died** | 8 (1) | 0 | - | - |
| **Ongoing pregnancy** | 36 (5) | | | |
| **Missing birth information** | 46 (6) | 0 | - | |
| **Pregnancy known completed** | 640 (89) | 694 (100) | - | |
| **Pregnancy loss before 24 weeks' gestation** | 16 (2) | 2 (<1) | NC | |
| **Mode of birth** | | | | |
| Pre-labor cesarean | 202 (32) | 124 (18) | 2.94 (2.23–3.87) | **2.58 (1.88–3.55)** |
| | | | p<0.001 | **p<0.001** |
| Cesarean after labor onset | 112 (18) | 77(11) | 2.62 (1.88–3.65) | **2.62 (1.79–3.85)** |
| | | | p<0.001 | **p<0.001** |
| Operative vaginal | 75 (12) | 71 (10) | 1.90 (1.33–2.73) | **2.14 (1.42–3.24)** |
| | | | p<0.001 | **p<0.001** |
| Unassisted vaginal | 233 (37) | 420 (61) | 1 (REF) | **1 (REF)** |
| Missing | 2 | 0 | - | |
| **Iatrogenic preterm birth <37 weeks'** | 87 (14) | 8 (1) | 13.94 (6.70–29.01) | **11.43 (5.07–25.75)** |
| | | | p<0.001 | **p<0.001** |
| **Spontaneous preterm birth <37 weeks'** | 27 (4) | 46 (7) | 0.64 (0.39–1.04) | **0.57 (0.32–1.01)** |
| | | | p = 0.072 | **p = 0.058** |
| | Infant outcomes (N = 634) | Infant outcomes (N = 705) | | |
| **Stillbirth** | 5 (1) | 2 (<1) | 2.80 (0.54–14.48) | **3.20 (0.54–19.07)** |
| | | | p = 0.220 | **p = 0.203** |
| **Live birth** | 627 (99) | 703 (100) | 0.36 (0.07–1.84) | **0.31 (0.05–1.87)** |
| | | | p = 0.219 | **p = 0.202** |
| **Neonatal unit admission** | 121 (19) | 37 (5) | 3.45 (2.39–4.97) | **3.08 (1.99–4.77)** |
| | | | p<0.001 | **p<0.001** |
| **Neonatal death** | 2 (<1) | 1 (<1) | 2.26 (0.20–25.00) | **3.91 (0.23–67.29)** |
| | | | p = 0.507 | **p = 0.348** |
| **Gestation at birth (weeks)*** | | | | |
| 22–27 | 6 (1) | 6 (1) | 1.27 (0.41–3.96) | **0.70 (0.15–3.16)** |
| | | | p = 0.680 | **p = 0.644** |
| 28–31 | 24 (4) | 6 (1) | 5.08 (2.06–12.53) | **3.98 (1.48–10.70)** |
| | | | p<0.001 | **p = 0.006** |
| 32–36 | 90 (14) | 51 (7) | 2.24 (1.56–3.22) | **1.87 (1.23–2.85)** |
| | | | p<0.001 | **p = 0.004** |
| 37 or more | 503 (81) | 639 (91) | 1 (REF) | **1 (REF)** |

*(Continued)*

**Table 3.** (Continued)

| Characteristic | Women with symptomatic SARS-CoV-2 (N = 722) | Historical comparison cohort (N = 694) | OR (95% CI) | aOR (95% CI)** |
|---|---|---|---|---|
| | Number (%) * | Number (%) * | | |
| Missing | 6 | 1 | - | - |

\* proportion of known.

\*\* adjusted for ethnicity, BMI, Any relevant previous medical problem and smoking.

\*\*\*excluding stillborn babies.

Of the 634 infants born to mothers with symptomatic SARS-CoV-2, 627 (99%) were live-born, 81% (n = 503) at term (Table 3). There were seven perinatal deaths in this group; five babies were stillborn and two died in the neonatal period, none of whom had confirmed SARS-CoV-2. This represents a perinatal mortality rate of 11 per 1000 births amongst hospitalized women with symptomatic SARS-CoV-2 (95% CI 4–23 per 1000). A total of 121 infants (19%) born to mothers with symptomatic SARS-CoV-2 were admitted to a neonatal unit compared to 5% (n = 37) of infants in the historical comparison cohort (aOR 3.08, 95% CI 1.99–4.77). Infants born to hospitalized mothers with symptomatic SARS-CoV-2 were more likely to be born at less than 37 weeks' and less than 32 weeks' of gestation compared to babies born to the historical comparison cohort of mothers without SARS-CoV-2 (aOR 1.87, 95% CI 1.23–2.85 and aOR 3.98, 95% CI 1.48–10.70 respectively) (Table 3). Infant outcomes were similar when comparing the overall group of women admitted to hospital with SARS-CoV-2 (both symptomatic and asymptomatic) to women without SARS-CoV2 (S8 Table).

More than one in four women that were symptomatic with SARS-CoV-2 were discharged prior to giving birth and have now completed their pregnancy (n = 206, 29%). In this group the majority went on to have liveborn infants (n = 204, 99%), at term (n = 186, 90% vs n = 314, 72% of those that gave birth at the time of admission) with a lower proportion requiring neonatal unit admission compared to those that were born at the time of admission (n = 17, 8% vs. n = 99, 23%).

There was no significant difference in the risk of stillbirth or neonatal death based on symptom status or in asymptomatic SARS-CoV-2 compared to the historical comparison cohort. The risk of infant admission to a neonatal unit was more than doubled in symptomatic compared to asymptomatic women with SARS-CoV-2 on hospital admission (19% vs 9%; OR 1.93, 95% CI 1.34–2.78, Table 4). Although the risk of neonatal unit admission was still increased when comparing women with asymptomatic SARS-CoV-2 to the historical comparison cohort (9% vs. 5%, aOR 1.84, 95% CI 1.12–3.03, Table 5). During this study period, universal screening was recommended for all babies born to mothers with confirmed SARS-CoV-2 that were admitted to neonatal units for specialist care from April 27[th], 2020. Babies born to mothers with SARS-CoV-2 that were not admitted to neonatal units were not routinely tested. Only 2% (n = 23) of infants tested positive for SARS-CoV-2 RNA, 12 within the first 12 hours of life. Five of the infants reported to have a positive test within 12 hours were admitted to a neonatal unit; only 2 of these infants had confirmed infection on re-testing. Babies born to women symptomatic of SARS-CoV-2 were more likely to be preterm compared to those born to asymptomatic women (19% vs 9%, aOR 1.88, 95% CI 1.20–2.95, Table 4), whose risk was not increased compared to the historical comparison cohort without SARS-CoV-2 (9% vs. 9%; aOR 1.17, 95% CI 0.75–1.83, Table 5).

**Table 4. Pregnancy and infant outcomes for symptomatic versus asymptomatic SARS-CoV-2 in pregnant women hospitalized in the UK.**

| Characteristic | Women with symptomatic SARS-CoV-2 (N = 722) | Women with asymptomatic SARS-CoV-2 (N = 426) | OR (95% CI) |
|---|---|---|---|
| | Number (%) * | Number (%) * | |
| Ongoing pregnancy | 36 (5) | 11 (3) | - |
| Missing birth information | 46 (6) | 19 (4) | - |
| Pregnancy known completed | 640 (89) | 396 (93) | - |
| Pregnancy loss before 24 weeks' gestation | 16 (2) | 15 (3) | |
| Mode of birth | | | |
| Pre-labor Cesarean | 202 (32) | 100 (26) | 1.51 (1.11–2.06) |
| | | | p = 0.009 |
| Cesarean after labor onset | 112 (18) | 53 (14) | 1.58 (1.08–2.31) |
| | | | p = 0.019 |
| Operative vaginal | 75 (12) | 54 (14) | 1.04 (0.69–1.55) |
| | | | p = 0.858 |
| Unassisted vaginal | 233 (37) | 174 (46) | 1 (REF) |
| Missing | 2 | 0 | - |
| | Infant outcomes (N = 634) | Infant outcomes (N = 385) | |
| Stillbirth | 5 (1) | 4 (1) | 0.76 (0.20–2.85) |
| | | | p = 0.683 |
| Live birth | 627 (99) | 381 (99) | - |
| Neonatal unit admission | 121 (19) | 35 (9) | 1.93 (1.34–2.78) |
| | | | p<0.000 |
| Neonatal death | 2 (<1) | 2 (1) | 0.61 (0.08–4.32) |
| | | | p = 0.617 |
| Gestation at birth (weeks)* | | | |
| 22–27 | 6 (1) | 3 (1) | 1.37 (0.34–5.51) |
| | | | p = 0.669 |
| 28–31 | 24 (4) | 1 (<1) | 16.4 (2.21–121.89) |
| | | | p = 0.006 |
| 32–36 | 90 (14) | 32 (8) | 1.92 (1.26–2.95) |
| | | | p = 0.003 |
| 37 or more | 503 (81) | 344 (91) | 1 |
| Median (IQR) | 39 (37–40) | 39 (39–40) | - |
| Missing | 6 | 1 | |

* excluding stillborn babies

## Discussion

This national prospective cohort study has reported an incidence of symptomatic SARS-CoV-2 in women hospitalized in pregnancy of 2.0 per 1000 maternities (95% CI 1.9–2.2) and an incidence of asymptomatic SAR-CoV-2 in women hospitalized in pregnancy of 1.2 per 1000 maternities (95% CI 1.1–1.4). Compared to hospitalized pregnant women without SARS-CoV-2, hospitalized women with symptomatic SARS-CoV-2 were more likely to be overweight or obese, to be of Black, Asian or Other minority ethnic group, and to have a relevant medical comorbidity including asthma and hypertension. The characteristics associated with asymptomatic SARS-CoV2 on hospital admission were Black, Asian or Other minority ethnicity and

**Table 5. Pregnancy and infant outcomes for pregnant women with asymptomatic confirmed SARS-CoV-2 infection hospitalized in the UK compared to a historical cohort without SARS-CoV-2 infection.**

| Characteristic | Women with asymptomatic SARS-CoV-2 (N = 426) | Historical comparison cohort (N = 694) | OR (95% CI) | aOR (95% CI)** |
|---|---|---|---|---|
| | Number (%) * | Number (%) * | | |
| **Pre-eclampsia** | 5 (1) | 8 (1) | 1.02 (0.33–3.13) | **0.73 (0.21–2.52)** |
| | | | p = 0.975 | p = 0.616 |
| **Ongoing pregnancy** | 11 (3) | 0 | - | |
| **Missing birth information** | 19 (4) | 0 | - | |
| **Pregnancy known completed** | 396 (93) | 694 (100) | - | - |
| **Pregnancy loss before 24 weeks' gestation** | 15 (3) | 2 (<1) | NC | |
| **Mode of birth** | | | | |
| Pre-labor cesarean | 100 (26) | 124 (18) | 1.95 (1.32–2.67) | **2.26 (1.62–3.17)** |
| | | | p<0.001 | p<0.001 |
| Cesarean after labor onset | 53 (14) | 77(11) | 1.66 (1.12–2.46) | **1.67 (1.12–2.52)** |
| | | | p = 0.011 | P = 0.013 |
| Operative vaginal | 54 (14) | 71 (10) | 1.84 (1.24–2.73) | **2.09 (1.38–3.16)** |
| | | | p = 0.003 | p<0.001 |
| Unassisted vaginal | 174 (46) | 420 (61) | 1 | **1** |
| Missing | 0 | 0 | - | |
| | Infant outcomes (N = 385) | Infant outcomes (N = 705) | | |
| **Stillbirth** | 4 (1) | 2 (<1) | 3.69 (0.67–20.21) | **3.63 (0.64–20.63)** |
| | | | p = 0.133 | p = 0.084 |
| **Live birth** | 381 (99) | 703 (100) | 0.27 (0.05–1.49) | **0.28 (0.05–1.57)** |
| | | | p = 0.133 | p = 0.146 |
| **Neonatal unit admission** | 35 (9) | 37 (5) | 1.82 (1.12–2.94) | **1.84 (1.12–3.03)** |
| | | | p = 0.015 | p = 0.017 |
| **Neonatal death** | 2 (1) | 1 (<1) | 3.73 (0.34–41.26) | **6.52 (0.58–73.13)** |
| | | | p = 0.283 | p = 0.128 |
| **Gestation at birth (weeks)*** | | | | |
| 22–27 | 3 (1) | 6 (1) | 0.93 (0.23–3.74) | **0.92 (0.18–4.64)** |
| | | | p = 0.917 | p = 0.920 |
| 28–31 | 1 (<1) | 6 (1) | 0.31 (0.04–2.58) | **0.35 (0.04–2.99)** |
| | | | p = 0.279 | p = 0.337 |
| 32–36 | 32 (8) | 51 (7) | 1.17 (0.74–1.85) | **1.30 (0.91–2.08)** |
| | | | p = 0.515 | p = 0.280 |
| 37 or more | 344 (91) | 639 (91) | 1 | **1** |
| Missing | 1 | 1 | - | - |

\* Percentages of those with complete data.

** adjusted for ethnicity, age and gestational diabetes.

***excluding stillborn babies.

gestational diabetes. Hospitalized pregnant women with symptomatic SARS-CoV-2 were more likely to be admitted to intensive care. They were more likely to have a cesarean or an operative vaginal birth, regardless of their symptom status, although the risk was greatest in those symptomatic for SARS-CoV-2. Hospitalized women with SARS-CoV-2 were more likely to have a preterm birth. This was driven by increased iatrogenic birth in women that were

symptomatic of SARS-CoV-2, as risk of preterm birth in asymptomatic women was not increased. Babies born to women with SARS-CoV-2 were more likely to be admitted to a neonatal unit, regardless of the mother's symptom status. There was no significant increase in stillbirth and neonatal deaths in hospitalized pregnant women with SARS-CoV-2 compared to those without SARS-CoV-2, or between those with symptoms and those that were asymptomatic but numbers in these groups were small.

This established method of nationwide, prospective case identification allowed rapid inception of reporting of population-based data, which has added robust confirmation to existing reports of the characteristics and outcomes for SARS-CoV-2 in women hospitalized in pregnancy. As UKOSS is the only national research platform for obstetrics in the UK, all other reports on this outcome during this time frame will be a subset of these data, including our initial publication of the first six weeks of the pandemic [3]. The comparison to a historical unexposed group without SARS-CoV-2 allowed conclusions to be drawn about the characteristics associated with hospitalization. However, we were unable to evaluate the outcomes in women with mild symptoms who were not admitted to hospital, nor the incidence and outcomes of asymptomatic infection in pregnant women not presenting to hospital for another cause. This study was undertaken in a high resource setting with universal health care at the point of access, therefore results are generalizable to similar settings.

The analysis of women by symptom status is a strength of this study. We have shown that since universal testing was recommended nationally, 64% of women admitted to hospital with confirmed SARS-CoV-2 were asymptomatic. To the best of our knowledge this is the first population-based study to report on symptom status in pregnancy. This analysis is important because women requiring hospital admission are by nature more likely to be at increased risk of adverse pregnancy outcome. For example, 3% (n = 31) of hospitalized pregnant women with SARS-CoV-2 had a pregnancy loss compared to <1% (n = 2) of those without SARS-CoV-2, but this could be a result of increased testing on admission in women presenting to hospital with symptoms of pregnancy loss as opposed to an effect of SARS-CoV-2 itself. This is supported by the finding of similar proportions of women with pregnancy loss in the analysis by symptom status. The small increase in the proportion of stillbirths in women with SARS-CoV-2 compared to those without SARS-CoV-2 may also be a result of this measurement bias. However, it is a limitation that despite national guidance, practice around initiation of universal screening will have varied between hospitals depending on testing capacity with some initiating screening earlier or later than recommended. This will have influenced the proportion of asymptomatic women detected throughout this study. Additionally, most symptoms of SARS-CoV-2 are non-specific and therefore the possibility of an alternative cause of these symptoms cannot be excluded.

We have confirmed that pregnant women hospitalized with SARS-CoV-2 were more likely to be Black, Asian or Other minority ethnicity, irrespective of symptom status, age, BMI and medical comorbidities. In the non-pregnant population, a recent systematic review has demonstrated that Black and Asian ethnic groups are more likely to be infected with SARS-CoV-2 compared to those of White ethnicity [17]. This suggests that the disproportionate impact could be attributable to increased infection in these ethnic groups. However, disparities in maternal mortality between ethnic groups are well known [13] and likely a result of complex interrelated factors, which may also be important in explaining the increased risk of SARS-CoV-2. For example, evidence suggests that people from ethnic minority backgrounds are more likely to live in larger household sizes [18], be of lower socioeconomic status [19], be employed as a public-facing key worker or less able to work from home [20], alongside other structural inequities. Further research is required to determine the reasons for this disparity in the risk of hospitalization with SARS-CoV-2 and how to mitigate the risk through the care we

provide. Women at increased risk should be informed of when and how to seek care and clinicians should use a lower threshold for investigation and management [21].

Current smoking was negatively associated with admission to hospital with symptomatic SARS-CoV-2. It is possible this is a result of residual confounding of ethnicity and/or geographical region in which women live. For example, data from the non-pregnant population suggest that there are substantial differences in smoking rates depending on country of birth (for example the highest proportion of current smokers are in those born in Poland (24.5%) and the lowest proportion in people born in India (5.3%) [22]) who could be over or underrepresented in the analysis by ethnic groups. Alternatively, rates of smoking in pregnancy vary across the UK (from 1.6% in Wokingham to 27.8% in Blackpool [23]) so it is plausible this could represent measurement bias, if places with lower smoking rates had higher rates of admission with SARS-CoV-2. Plausible biological explanations also exist, for example, a nicotine may exert a potential anti-inflammatory effect and inhibit the production of pro-inflammatory cytokines, alternatively increased nitric oxide may inhibit the replication of SARS-CoV-2 at cell entry [24].

Whilst we have demonstrated that the absolute risk of poor maternal outcome in hospitalized women with SARS-CoV-2 between 1st March and 31st August was low, data from the intensive care national audit and research centre (ICNARC) suggest that the proportion of women admitted to intensive care that are currently or recently pregnant may be increasing [25]. The reason for the increasing proportion of severe SARS-CoV-2 in pregnancy compared to the general population is not known and further research is required. In this study we have identified that very few pregnant women (2%) were treated with anti-viral medications, much lower than in some other high resource countries such as Italy, where nearly a quarter of women (23%) received treatment with anti-viral agents [26]. Evidence about pharmacological management in the general population is improving [16]. We have identified risk factors that increase the likelihood of women being hospitalized with SARS-CoV-2 and therefore potentially those that would gain the most from evidence-based treatment. The RCOG recommends that clinicians should consider the use of medications which have been shown to be beneficial and that pregnant women should be offered inclusion in trials of therapy to reduce the severity of SARS-CoV-2 [21].

We have demonstrated that hospitalization with SARS-CoV-2 is associated with increased risk of cesarean section, irrespective of symptom status. This supports recent systematic reviews, which have reported up to 60% of pregnant women with SARS-CoV-2 had a cesarean birth [2, 27]. We also provide clear data on the indication for interventional birth, the majority of which were unrelated to SARS-CoV-2. The finding that cesareans are increased irrespective of symptom status suggests that some of the overall increased risk is a result of measurement bias, as women with pregnancy complications requiring cesarean birth are more likely to present to hospital and be screened and delivered than the comparison population. However, changes in maternity practice may also contribute to this increase and this warrants further investigation, as it has implications for informed decision-making, future pregnancies and resource availability in both high and low-income settings.

Nearly one in five (n = 120) infants of hospitalized mothers with SARS-CoV2 were born preterm and the majority of these preterm infants required neonatal care (n = 98, 63%). However, this was driven by iatrogenic preterm birth and there was a suggestion that spontaneous preterm birth was reduced. A number of other studies undertaken in high-resource settings have also reported a significant reduction in preterm births and hypothesized that this was due to beneficial effects of the pandemic control measures such as handwashing and social distancing reducing infections, reduced air pollution and greater physical rest [28, 29]. Further research is required to determine the impact of social and behavioral changes in differing risk

groups and their impact on maternity outcomes in this pandemic. Additionally, the high rates of iatrogenic preterm birth and neonatal unit admission due to prematurity continue to suggest that the indirect effects of SARS-CoV-2 on delivery of maternity care are important and will continue to be so, especially in the absence of vaccine use in pregnant women [30]. Our data suggest that clinicians should be reassured that women with mild SARS-CoV-2 can be discharged from hospital to continue their pregnancy safely.

## Conclusion

This national study has demonstrated that raised BMI, Black, Asian or Other minority ethnicity and relevant medical co-morbidities are associated with hospitalization with symptomatic SARS-CoV-2. These groups should be considered for inclusion and prioritization when testing SARS-CoV-2 vaccine efficacy and safety. Risks could be minimized where possible through pre-pregnancy optimization of weight and medical co-morbidities. Further research is required to determine why women of Black, Asian and Other minority ethnicity are disproportionately affected and how to minimize the impact of this through care provision. Overall, just under one in 10 women admitted to hospital with symptoms of SARS-CoV-2 in pregnancy required critical care. Outcomes in this group are predominantly good, but women should be treated with medications known to be effective, to ensure their outcomes improve in line with the general population. The proportion of cesarean births and iatrogenic preterm birth is high which provides clear evidence of the indirect impact of SARS-CoV-2 on maternity care in a high-income setting. This needs to be taken into account in guidance as the pandemic continues and as SARS-CoV-2 moves to become an endemic infection, in order to prevent immediate complications such as neonatal prematurity and long-term complications associated with over-intervention in care.

## Supporting information

**S1 Table. Relevant comorbidities.**
(DOCX)

**S2 Table. UK Census coding for ethnic group.**
(DOCX)

**S3 Table. Sensitivity analysis for pre-existing medical comorbidities in women with symptomatic SARS-CoV-2 infection.**
(DOCX)

**S4 Table. Characteristics of pregnant women with confirmed SARS-CoV-2 infection admitted to hospital in the UK compared to a historical cohort without SARS-CoV-2.**
(DOCX)

**S5 Table. Characteristics of pregnant women with symptomatic versus asymptomatic SARS-CoV-2 hospitalized in the UK.**
(DOCX)

**S6 Table. Maternal and perinatal outcomes and diagnoses amongst women with confirmed SARS-CoV-2 infection in pregnancy admitted to intensive care.**
(DOCX)

**S7 Table. Indication for cesarean in women with confirmed symptomatic, asymptomatic SARS-CoV-2 and a historical comparison cohort.**
(DOCX)

**S8 Table. Hospital, pregnancy and infant outcomes amongst women with confirmed SARS-CoV-2 infection in pregnancy compared to a historical cohort without SARS-CoV-2 infection.**
(DOCX)

## Acknowledgments

The authors would like to acknowledge the assistance of UKOSS reporting clinicians, the UKOSS Steering Committee and The NIHR Clinical Research Networks without whose support this research would not have been possible.

## Author Contributions

**Conceptualization:** Edward Morris, Patrick O'Brien, Maria Quigley, Peter Brocklehurst, Jennifer J. Kurinczuk, Marian Knight.

**Data curation:** Marian Knight.

**Formal analysis:** Nicola Vousden, Marian Knight.

**Funding acquisition:** Patrick O'Brien, Maria Quigley, Peter Brocklehurst, Jennifer J. Kurinczuk, Marian Knight.

**Investigation:** Nicola Vousden, Kathryn Bunch, Marian Knight.

**Methodology:** Nicola Vousden, Kathryn Bunch, Marian Knight.

**Project administration:** Marian Knight.

**Supervision:** Kathryn Bunch, Marian Knight.

**Writing – original draft:** Nicola Vousden.

**Writing – review & editing:** Nicola Vousden, Kathryn Bunch, Edward Morris, Nigel Simpson, Christopher Gale, Patrick O'Brien, Maria Quigley, Peter Brocklehurst, Jennifer J. Kurinczuk, Marian Knight.

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
