## [Decision Letter · Decision Letter 0]

10 Mar 2021

PONE-D-21-00575

The incidence, characteristics and outcomes of pregnant women hospitalized with symptomatic and asymptomatic SARS-CoV-2 infection in the UK from March to September 2020: a national cohort study using the UK Obstetric Surveillance System (UKOSS)

PLOS ONE

Dear Prof Knight,

Thank you for submitting your manuscript to PLOS ONE. After careful consideration, we feel that it has merit but does not fully meet PLOS ONE’s publication criteria as it currently stands. Therefore, we invite you to submit a revised version of the manuscript that addresses the points raised during the review process.

We look forward to receiving your revised manuscript.

Kind regards,

Diane Farrar

Academic Editor

PLOS ONE

Journal Requirements:

2. Thank you for stating in your ethics statement in the online submission form "Anonymised data were collected and therefore individual participant consent was not required." Please clarify whether all data were fully anonymized before you accessed them and/or whether the IRB or ethics committee waived the requirement for informed consent.

3.We note that you have indicated that data from this study are available upon request. PLOS only allows data to be available upon request if there are legal or ethical restrictions on sharing data publicly. For information on unacceptable data access restrictions, please see http://journals.plos.org/plosone/s/data-availability#loc-unacceptable-data-access-restrictions.

4.Thank you for stating the following in the Competing Interests section:

"MK, MQ, PB, PO’B, JJK received grants from the NIHR in relation to the submitted work. KB, NV, NS, CG have no conflicts of interest to declare. EM is Trustee of RCOG, British Menopause Society and Newly Chair of the Board of Trustees Group B Strep Support."

Additional Editor Comments:

Please spell out in full in the first instance MBRRACE-UK

Please check for typos ie ‘inflation actors’ inflation factors…….. line 166

Please define ethnic groups (as pointed out by reviewer 1, but also for Black ethnicity)

Line 249-250 Smoking Babies?

Line 394-396 check sentence structure it doesn’t seem to make sense

Reviewers' comments:

Reviewer's Responses to Questions

**Comments to the Author**

1. Is the manuscript technically sound, and do the data support the conclusions?

Reviewer #1: Yes

Reviewer #2: Yes

2. Has the statistical analysis been performed appropriately and rigorously? 

Reviewer #1: I Don't Know

Reviewer #2: Yes

3. Have the authors made all data underlying the findings in their manuscript fully available?

Reviewer #1: Yes

Reviewer #2: No

4. Is the manuscript presented in an intelligible fashion and written in standard English?

Reviewer #1: Yes

Reviewer #2: Yes

5. Review Comments to the Author

Reviewer #1: This is a well written and excellent study. As a US reviewer I would like to note that the description of Asian needs further definition as it may differ between the US and UK. Asian in the USA includes Chinese ethnicity but on Table 1 of the manuscript there is both Asian and Chinese listing. Does that mean Asian in the UK is only Southeast Asian/Indian?

The statement on line 211 that htn increases risk of admission does not appear to be supported by the small numbers noted on table 2. Perhaps a further explanation of the authors conclusion on this point would be helpful to the reader.

line 255 when noting symptomatic SARS and pregnancy loss less than 24 weeks , if the authors do not wish to add the historical comparison can they explain why that is not used.

A further limitation to consider is that the "symptomatic" SARS CoV2 (other than the pathognomonic cxr for pneumonia) may have been symptomatic from some other etiology other than CoV2 .It is not clear if other causes were ruled out before deciding that Cov2 was the reason for the symptoms. If this was done it needs to be so stated.

Reviewer #2: This study uses data from an obstetric surveillance system in the United Kingdom to study occurrence and selected outcomes associated with SARS-CoV-2 infection. Data on various characteristics of pregnant women and their births are presented, contributing useful information on this novel virus. The study population does not include all pregnant women in the catchment area, unfortunately, and bias arising from reverse causation is possible in which women at highest risk of adverse outcomes may be more likely to be hospitalized and tested for the virus. The authors are aware of this possibility and discuss it appropriately. A major strength of this study is inclusion of women with both asymptomatic and symptomatic infection. Comparisons of these two groups of women helps address the possibility of reverse causation bias.

Study methods are sound. There is an over-reliance on statistical testing, however. For example, the authors used statistical testing with stepwise forward selection to determine which factors to control in multivariable logistic regression models. However, odds ratios can be confounded by factors that are associated with the outcome at higher (less extreme) p-values. Another example is the odds ratio for spontaneous preterm birth and symptomatic infection. The adjusted point estimate (0.57, 95% CI 0.32-1.01) should not be interpreted as no association. P-values could be dropped from all tables.

It is not clear why adjusted odds ratios were omitted for some comparisons. The reasoning should be described in the methods section.

There is a large amount of repeated information across tables. For example, the distribution of factors for the historical cohort is repeated in tables 1 and 2. There is even more redundancy across tables 3, 4, and 5. Tables could be combined for efficiency of space and to help the reader compare differences between symptomatic and asymptomatic women.

There is no discussion of the findings on maternal smoking. Why might cigarette smoking reduce the risk of SARS-CoV-2 infection? The authors can draw from the literature for non-pregnant populations in their discussion of smoking.

It would be helpful for the authors to discuss findings related to specific illnesses that were associated with

In the last paragraph of the introduction, the sentence containing “….risk of infection with SARS-CoV-2 or resulting morbidity (e.g., diabetes) to explore if any were…” suggests that the authors were investigating diabetes as a resulting morbidity.

In both the introduction and the discussion sections, the authors describe their study results as unbiased. There may indeed be a bias in the reported data if getting to hospital and being tested is associated with women’s characteristics and underlying risk of exposure to the virus.

With respect to this potential bias, it would be useful for the authors to discuss their findings for pre-existing conditions. Differences in results between symptomatic and asymptomatic women are interesting.

The authors discuss their findings for ethnicity and outcomes by commenting that disparities likely result from a complexity of interrelated factors. They provide examples of such factors. It is important to frame this discussion in a way that removes ‘fault’ from individuals. Persons with the highest rates of infection don’t live with more people or work certain jobs necessarily by choice; most often living situations result from societal forces rather than from individuals choosing their circumstances.

6. PLOS authors have the option to publish the peer review history of their article (what does this mean?). If published, this will include your full peer review and any attached files.

Reviewer #1: **Yes: **kay daniels

Reviewer #2: No

---

## [Author Response · Author response to Decision Letter 0]

15 Apr 2021

Editor comments

Journal Requirements:

The reference list has been checked and is complete. 

 These have been reviewed and amended as required.

2. Thank you for stating in your ethics statement in the online submission form "Anonymised data were collected and therefore individual participant consent was not required." Please clarify whether all data were fully anonymized before you accessed them and/or whether the IRB or ethics committee waived the requirement for informed consent.

All data were fully anonymised before we received them and the ethics committee waived the requirement for informed consent on this basis.

3.We note that you have indicated that data from this study are available upon request. PLOS only allows data to be available upon request if there are legal or ethical restrictions on sharing data publicly. For information on unacceptable data access restrictions, please see http://journals.plos.org/plosone/s/data-availability#loc-unacceptable-data-access-restrictions.

There are ethics committee restrictions on data availability and a statement has been added to the paper to clarify at line 147-150: 

“Data cannot be shared publicly because of confidentiality issues and potential identifiability of sensitive data as identified within the Research Ethics Committee application/approval. Requests to access the data can be made by contacting the National Perinatal Epidemiology Unit data access committee via general@npeu.ox.ac.uk.”

b) If there are no restrictions, please upload the minimal anonymized data set necessary to replicate your study findings as either Supporting Information files or to a stable, public repository and provide us with the relevant URLs, DOIs, or accession numbers. Please see http://www.bmj.com/content/340/bmj.c181.long for guidelines on how to de-identify and prepare clinical data for publication. For a list of acceptable repositories, please seehttp://journals.plos.org/plosone/s/data-availability#loc-recommended-repositories.

The cover letter and the paper have been updated to clarify that there are ethics committee restrictions on data availability at line 147-150: 

“Data cannot be shared publicly because of confidentiality issues and potential identifiability of sensitive data as identified within the Research Ethics Committee application/approval. Requests to access the data can be made by contacting the National Perinatal Epidemiology Unit data access committee via general@npeu.ox.ac.uk.”

4.Thank you for stating the following in the Competing Interests section:

"MK, MQ, PB, PO’B, JJK received grants from the NIHR in relation to the submitted work. KB, NV, NS, CG have no conflicts of interest to declare. EM is Trustee of RCOG, British Menopause Society and Newly Chair of the Board of Trustees Group B Strep Support."

We have amended the cover letter to state that the competing interests do not alter our adherence to PLOS ONE policies on sharing data and materials. However, data cannot be shared publicly because of confidentiality issues and potential identifiability of sensitive data as identified within the Research Ethics Committee application/approval. Contact details for data access requests have also been provided. 

Additional Editor Comments:

Please spell out in full in the first instance MBRRACE-UK

This has been added at line 116-7.

Please check for typos ie ‘inflation actors’ inflation factors…….. line 166

This has been amended and the document searched for any other errors. 

Please define ethnic groups (as pointed out by reviewer 1, but also for Black ethnicity)

This has been clarified by the addition of the following at line 109-111 in the methods section and a supplementary table added to supporting information detailing the classification of these groups:

“Women’s ethnic groups were identified on maternal self-report as recorded in the maternity records, and were classified based on the census classification for England and Wales (S2 Table).”

Line 249-250 Smoking Babies?

We apologise for this formatting error which has been corrected

Line 394-396 check sentence structure it doesn’t seem to make sense

This sentence has been clarified as follows:

“We have identified risk factors that increase the likelihood of women being hospitalized with SARS-CoV-2 and therefore potentially those that would gain the most from evidence-based treatment.”

Reviewer comments to the Author

Reviewer #1: This is a well written and excellent study. As a US reviewer I would like to note that the description of Asian needs further definition as it may differ between the US and UK. Asian in the USA includes Chinese ethnicity but on Table 1 of the manuscript there is both Asian and Chinese listing. Does that mean Asian in the UK is only Southeast Asian/Indian?

The following sentence has been added to the methods at line 109-111 to clarify the classification of ethnicity in this paper and a supplementary table has been added as shown in the editor’s comments. It is usual practice in the UK to combine Chinese ethnicity with ‘any other ethnicity’ but for the purpose of this analysis they have been included separately as there was a theoretical reason, based on the assumed origin of SARS-CoV-2 in Wuhan, for women of Chinese ethnicity to be at greater risk of infection. 

 “Women’s ethnic groups were identified on maternal self-report as recorded in the maternity records and were classified based on the census classification for England and Wales (S2 Table). 

The statement on line 211 that htn increases risk of admission does not appear to be supported by the small numbers noted on table 2. Perhaps a further explanation of the authors conclusion on this point would be helpful to the reader.

The sentence has been amended by pointing out that numbers are small and the aOR from the sensitivity analysis have been added so that the readers can see the association for hypertension is of borderline significance: 

Line 222-223: “In the sensitivity analysis, there was some evidence that asthma and hypertension specifically increased the risk of admission, although the numbers in some groups were small (aOR 2.12, 95% CI 1.25-3.58 and aOR 3.63, 95% CI 0.99 – 13.30 respectively, Table S3).”

line 255 when noting symptomatic SARS and pregnancy loss less than 24 weeks , if the authors do not wish to add the historical comparison can they explain why that is not used.

This is already explained in the discussion in lines 273-275 but the following sentence has been added to the methods to highlight this decision.

“Early pregnancy outcomes were not compared with the historical comparison group due to the risk of measurement bias resulting from screening for SARS-CoV-2 on admission to hospital with symptoms of pregnancy loss.”

A further limitation to consider is that the "symptomatic" SARS CoV2 (other than the pathognomonic cxr for pneumonia) may have been symptomatic from some other etiology other than CoV2 .It is not clear if other causes were ruled out before deciding that Cov2 was the reason for the symptoms. If this was done it needs to be so stated.

The following statement has been added to Line 396-398 of the discussion to highlight this limitation:

“Additionally, most symptoms of SARS-CoV-2 are non-specific and therefore the possibility of an alternative cause of these symptoms cannot be excluded.” 

Reviewer #2: This study uses data from an obstetric surveillance system in the United Kingdom to study occurrence and selected outcomes associated with SARS-CoV-2 infection. Data on various characteristics of pregnant women and their births are presented, contributing useful information on this novel virus. The study population does not include all pregnant women in the catchment area, unfortunately, and bias arising from reverse causation is possible in which women at highest risk of adverse outcomes may be more likely to be hospitalized and tested for the virus. The authors are aware of this possibility and discuss it appropriately. A major strength of this study is inclusion of women with both asymptomatic and symptomatic infection. Comparisons of these two groups of women helps address the possibility of reverse causation bias.

Study methods are sound. There is an over-reliance on statistical testing, however. For example, the authors used statistical testing with stepwise forward selection to determine which factors to control in multivariable logistic regression models. However, odds ratios can be confounded by factors that are associated with the outcome at higher (less extreme) p-values. Another example is the odds ratio for spontaneous preterm birth and symptomatic infection. The adjusted point estimate (0.57, 95% CI 0.32-1.01) should not be interpreted as no association. 

We have clarified the wording of the results in line 291-292 as follows: “Whilst there was an apparently lower proportion of spontaneous preterm births amongst women admitted with symptomatic SARS-CoV-2, this was partly explained by confounding (4% vs. 7%, aOR 0.57, 95% CI 0.32-1.01, Table 3).”

We hope the existing wording of the discussion, Line 455-457 is satisfactory: “However, this was driven by iatrogenic preterm birth and there was a suggestion that spontaneous preterm birth was reduced”. 

Other areas where we feel we have demonstrated appropriate judgement of significance is the inclusion of maternal age in our multivariate model based on our a priori hypothesis, and presentation of other ‘borderline significant’ outcomes, for example the inclusion of hypertension as a risk factor for admission (aOR 3.63, 95% CI 0.99 – 13.30).

P-values could be dropped from all tables.

P-values were added on the request of the PLOS One administrative team on submission of the paper, however we are happy to remove them if the editors would prefer. 

It is not clear why adjusted odds ratios were omitted for some comparisons. The reasoning should be described in the methods section.

The following sentence in the method line 164 - 166 has been clarified: “Any further potential confounders identified as significantly associated during the univariable analysis were adjusted in the multivariable unconditional regression analysis.”

There is a large amount of repeated information across tables. For example, the distribution of factors for the historical cohort is repeated in tables 1 and 2. There is even more redundancy across tables 3, 4, and 5. Tables could be combined for efficiency of space and to help the reader compare differences between symptomatic and asymptomatic women.

Combining the tables would result in very wide tables (9 columns minimum) and we feel this would not improve the clarity of the content, however we can make this change if the editors feel this would be preferable. 

There is no discussion of the findings on maternal smoking. Why might cigarette smoking reduce the risk of SARS-CoV-2 infection? The authors can draw from the literature for non-pregnant populations in their discussion of smoking.

We thank the reviewer for highlighting this area of discussion. The following has been added to the discussion at line 415-425: “Current smoking was negatively associated with admission to hospital with symptomatic SARS-CoV-2. It is possible this is a result of residual confounding of ethnicity and/or geographical region in which women live. For example, data from the non-pregnant population suggest that there are substantial differences in smoking rates depending on country of birth (for example the highest proportion of current smokers are in those born in Poland (24.5%) and the lowest proportion in people born in India (5.3%)[22]) who could be over or underrepresented in the analysis by ethnic groups. Alternatively, rates of smoking in pregnancy vary across the UK (from 1.6% in Wokingham to 27.8% in Blackpool[23]) so it is plausible this could represent measurement bias, if places with lower smoking rates had higher rates of admission with SARS-CoV-2. Plausible biological explanations also exist, for example, nicotine may exert an anti-inflammatory effect and inhibit the production of pro-inflammatory cytokines, alternatively increased nitric oxide may inhibit the replication of SARS-CoV-2 at cell entry[24].” 

In the last paragraph of the introduction, the sentence containing “….risk of infection with SARS-CoV-2 or resulting morbidity (e.g., diabetes) to explore if any were…” suggests that the authors were investigating diabetes as a resulting morbidity.

This has been reworded as follows:

Line 171-2: “A sensitivity analysis was undertaken to explore if any pre-existing medical problems (e.g. diabetes), that might have increased the risk of infection with SARS-CoV-2 or resulting morbidity, were independently associated with the outcome.” 

In both the introduction and the discussion sections, the authors describe their study results as unbiased. There may indeed be a bias in the reported data if getting to hospital and being tested is associated with women’s characteristics and underlying risk of exposure to the virus.

This measurement bias has already been explored in the discussion in lines 385-393. The wording in the discussion has been altered to remove the word ‘unbiased’. The use of the word ‘unbiased’ in the introduction is in description of the available evidence and therefore has not been changed: 

“However, the majority of studies to date are case reports, case series and institutional or registry non-population-based cohort studies and there is a lack of population-level data to inform accurate incidence rates and unbiased descriptions of characteristics and outcomes.” 

With respect to this potential bias, it would be useful for the authors discuss their findings for pre-existing conditions. Differences in results between symptomatic and asymptomatic women are interesting.

The sentence that describes these differences in the results has been expanded:

Line 212-213: “In the sensitivity analysis, there was some evidence that asthma and hypertension specifically increased the risk of admission with symptomatic SARS-CoV-2, although the numbers in some groups were small (aOR 2.12, 95% CI 1.25-3.58 and aOR 3.63, 95% CI 0.99 – 13.30 respectively, Table S3”

The authors discuss their findings for ethnicity and outcomes by commenting that disparities likely result from a complexity of interrelated factors. They provide examples of such factors. It is important to frame this discussion in a way that removes ‘fault’ from individuals. Persons with the highest rates of infection don’t live with more people or work certain jobs necessarily by choice; most often living situations result from societal forces rather than from individuals choosing their circumstances.

This sentence has been added to as follows: 

Line 407-410: “For example, evidence suggests that people from ethnic minority backgrounds are more likely to live in larger household sizes[18], be of lower socioeconomic status [19], be employed as a public-facing key worker or less able to work from home [20], alongside other structural inequities.”

---

## [Editor Report · Decision Letter 1]

21 Apr 2021

The incidence, characteristics and outcomes of pregnant women hospitalized with symptomatic and asymptomatic SARS-CoV-2 infection in the UK from March to September 2020: a national cohort study using the UK Obstetric Surveillance System (UKOSS)

PONE-D-21-00575R1

Dear Prof. Knight,

We’re pleased to inform you that your manuscript has been judged scientifically suitable for publication and will be formally accepted for publication once it meets all outstanding technical requirements.

Kind regards,

Diane Farrar

Academic Editor

PLOS ONE

Additional Editor Comments:

As you will be aware there is a move away from using p values, as many results are not appropriately dichotomized in this way, however readers may use p values as a guide and some may have limited knowledge of the meaning of some estimates, therefore I would be happy for p values to remain in the tables. Also please leave the tables in their present format, I agree that amalgamating them would result in overly large tables. Would you consider changing 'pregnant mothers' to 'pregnant women', abstract and intro and throughout the manuscript when referring to antenatal women, both terms are used and one should be chosen for consistency to describe women before they have given birth, post birth, women can be referred to as mothers of course.

---

## [Editor Report · Acceptance letter]

28 Apr 2021

PONE-D-21-00575R1 

The incidence, characteristics and outcomes of pregnant women hospitalized with symptomatic and asymptomatic SARS-CoV-2 infection in the UK from March to September 2020: a national cohort study using the UK Obstetric Surveillance System (UKOSS) 

Dear Dr. Knight:

I'm pleased to inform you that your manuscript has been deemed suitable for publication in PLOS ONE. Congratulations! Your manuscript is now with our production department. 

Kind regards, 

on behalf of

Dr. Diane Farrar 

Academic Editor

PLOS ONE